# Flexible Temperature Sensor Utilizing MWCNT Doped PEG-PU Copolymer Nanocomposites

**DOI:** 10.3390/mi13020197

**Published:** 2022-01-27

**Authors:** Amit Kumar, Pen-Yi Hsieh, Muhammad Omar Shaikh, R. K. Rakesh Kumar, Cheng-Hsin Chuang

**Affiliations:** 1Institute of Medical Science and Technology, National Sun Yat-sen University, Kaohsiung 80424, Taiwan; amitkumar82829@gmail.com (A.K.); bernie82963724@gmail.com (P.-Y.H.); rakeshrk141996@gmail.com (R.K.R.K.); 2Sustainability Science and Engineering Program, Tunghai University, Taichung 407224, Taiwan; omar@thu.edu.tw

**Keywords:** temperature sensor, thermosensitive, MWCNT, copolymer nanocomposites

## Abstract

In this study, polyethylene glycol (PEG) and polyurethane (PU)-based shape-stabilized copolymer nanocomposites were synthesized and utilized for developing low-cost and flexible temperature sensors. PU was utilized as a flexible structural material for loading a thermosensitive phase change PEG polymer by means of physical mixing and chemical crosslinking. Furthermore, the introduction of multi-walled carbon nanotubes (MWCNT) as a conductive filler in the PEG-PU copolymer resulted in a nanocomposite with thermoresistive properties. MWCNT loading concentrations from 2 wt.% to 10 wt.% were investigated, to attain the optimum conductivity of the nanocomposite. Additionally, the effect of MWCNT loading concentration on the thermosensitive behavior of the nanocomposite was analyzed in the temperature range 25 °C to 50 °C. The thermosensitive properties of the physically mixed and crosslinked polymeric nanocomposites were compared by spin coating the respective nanocomposites on screen printed interdigitated (IDT) electrodes, to fabricate the temperature sensor. The chemically crosslinked MWCNT-PEG-PU polymeric nanocomposite showed an improved thermosensitive behavior in the range 25 °C to 50 °C, compared to the physically mixed nanocomposite. The detailed structural, morphological, thermal, and phase transition properties of the nanocomposites were investigated using XRD, FTIR, and DSC analysis. XRD and FTIR were used to analyze the crystallinity and PEG-PU bonding of the copolymer nanocomposite, respectively; while the dual phase (solid–liquid) transition of PEG was analyzed using DSC. The proposed nanocomposite-based flexible temperature sensor demonstrated excellent sensitivity, reliability and shows promise for a wide range of bio-robotic and healthcare applications.

## 1. Introduction

In recent years, significant research efforts have been made towards the advancement of smart materials and their properties. In particular, the development of smart nanocomposites, obtained by combining nanomaterials and polymers, is rapidly progressing, due to their enhanced sensing capabilities [1]. Owing to their excellent electrical and physical properties, carbon nanotubes (MWCNTs) have been extensively used as a conductive material for synthesis of polymeric nanocomposites [2,3,4,5]. MWCNT-based polymeric nanocomposite sensing materials have been used for various transduction applications including in gas, piezoresistive, mechanical, and biological sensors, among others [2,3,4,5,6]. However, only a limited number of research investigations have been conducted about the thermoresistive properties of MWCNT-based nanocomposite materials. Thermoresistivity (‘thermo’ and ‘resistivity’) refers to the rate of variation in the electrical resistance of the material, depending on temperature change, which is known as the temperature coefficient of resistance (TCR). The TCR of materials can be categorized as being either a positive temperature coefficient of resistance (PTC), where resistance increases with increase in temperature, or a negative temperature coefficient of resistance (NTC), where resistance and temperature are inversely related. Ebbesen et al. and others have reported that multi-walled carbon nanotubes (MWCNT) exhibit negative temperature coefficient of resistance (NTC) [7,8,9]. A NTC between −140 °C to 130 °C has been reported in the literature for 0.025 wt.% to 0.5 wt.% MWCNT loading concentrations [10].

Traditionally, various materials, including polymers, have been utilized as the structural material for the loading of solid–liquid phase change materials (PCMs). Various carbon nanomaterials, including graphene, MWCNT, graphene oxide [11,12,13], and a variety of polymers such as polyurethane [14] and poly (methyl-methacrylate) [15], have been used to encapsulate or load the phase change materials for preparation of smart materials. Commonly, the structural material is selected such that it has a higher melting point than the phase change material. This concept, combined with the flexibility of polyurethane, made it an ideal choice as the structural material used in this study for loading of PEG (phase change material).

A comparative investigation of the thermosensitive properties of the physically mixed PEG-PU polymers and a chemically crosslinked PEG-PU copolymer was performed, utilizing MWCNT as the conductive filler in the nanocomposite. We first optimized the MWCNT loading concentration from 2 wt.% to 10 wt.% in polyurethane. The effect of conductive filler loading on thermoresistive properties was observed and the MWCNT-PU composite showed a NTC (negative temperature coefficient) behavior in the temperature range 298–323 K, which agrees with previously reported data. Furthermore, PEG was used as a thermosensitive material, due to its phase change properties at physiological temperatures. Herein, a PEG-PU copolymer was prepared by two pathways: (1) physical mixing, and (2) chemical crosslinking. Crosslinked PU copolymer was synthesized by utilizing diphenylmethane diisocyanate (MDI), PEG, and 2-hydroxypropyl-β-cyclodextrin (HP-β-CD), while physically mixed PEG-PU polymers were obtained simply by mixing using a magnetic stirrer. While the addition of PEG in the nanocomposite enhanced the thermosensitive behavior of the nanocomposite, it was observed that the chemically crosslinked PEG-PU copolymer demonstrated better thermo-resistive behavior in the temperature range 298–323 K.

## 2. Experimental

### 2.1. Materials

Diphenylmethane diisocyanate (MDI), 2-Hydroxypropyl-β-cyclodextrin (HP-β-CD), PEG (MW = 1500), and N, N-dimethylformamide (DMF, AR) were purchased from Sigma Aldrich. MWCNTs (L = 5–15 µm, d = 10–20 nm) were purchased from Conjutek co. Ltd. (Taipei, Taiwan).

### 2.2. PU Copolymer Synthesis

Crosslinked PU copolymer network was synthesized using previously reported data [16,17]. Briefly, 6 mmol MDI solution in DMF was gradually injected in a 3 mmol PEG solution in DMF at 90 °C for 6 h, using a heating mantle in nitrogen atmosphere. A separate solution of 0.857 mmol (HP-β-CD) solution (in DMF) was prepared and injected in the solution; whereby, PEG acts as soft sector and HP-β-CD acts as chain extender in the copolymer. The solution was kept for 24 h in the same conditions under continuous stirring. Later, the solution was transferred to an oven for thermal curing for 3 days at 80 °C. The complete process for synthesizing crosslinking PU-PEG copolymer is shown in Figure 1a–d. Finally, the solution was centrifuged using ethanol, and a yellowish color was obtained after washing and drying the mixture solution.

### 2.3. MWCNT-PEG-PU Nanocomposite Synthesis and Sensor Fabrication

In order to optimize the conductive filler loading in the composite, the MWCNT concentration was varied from 2 wt.% to 10 wt.% based on our previous reported articles for MWCNT-nanocomposite [18]. Additionally, studies have shown that longer and thicker MWCNTs (with greater diameter) support the thermal conductivity, which is necessary to fabricate a temperature sensor. While the shorter and thinner nanowires have a higher thermal resistance [19]. Briefly, a homogeneous solution of polyurethane was prepared by mixing it in DMF and then the required amount of MWCNT (2–10 wt.%) was added to the solution. After making the composite, it was spin coated on a screen-printed electrode to investigate the effect of MWCNT loading concentration on electrical and thermoresistive properties. Furthermore, to prepare the MWCNT-PEG-PU nanocomposite, two distinct pathways were considered, which included (1) physical mixing and (2) chemical crosslinking of PEG and PU. For physical mixing, different ratios (PEG: PU = 1:1 to 1:5) of PEG and PU were tested. After assessing the phase change properties and DSC analysis, we found that PEG:PU composite in a 1:2 ratio showed the most promising thermosensitive properties. Hence this ratio was used for loading MWCNT to make a MWCNT-PEG-PU nanocomposite by physical mixing. A schematic for the physical mixing process is illustrated in Figure 1i–l. For chemical crosslinking, 1.5 g pre-synthesized crosslinking PEG-PU copolymer powder was mixed in THF under vigorous stirring at 80 °C for 2 h on a hot plate. Later, the required amount of MWCNTs were loaded to make the MWCNT-PEG-PU nanocomposite, as shown in Figure 1f. Finally, the nanocomposite was cured in an oven by evaporating the solvent at 80 °C. A schematic process for synthesis and fabrication of a crosslinked MWCNT-PEG-PU nanocomposite-based temperature sensor is shown in Figure 1i–h. Both physically mixed and chemically crosslinked MWCNT-PEG-PU nanocomposites were spin coated on IDT electrodes to test the thermoresistive properties of the prepared material.

## 3. Results and Discussion

Fourier transformed infrared (FTIR) analysis of the nanocomposite was performed using a Thermo Nicolet iS5. The FTIR spectra of the reactants, copolymers, and nanocomposite is shown in Figure 2a. The FTIR spectra of the PEG (MW = 1500 and 3000 g/mol) shows minor characteristic bands at 3428, 1288, and 1250 cm^−1^, which represent the O–H stretching and bending vibrations. From the FTIR spectra of pristine PEG, the characteristic peaks at 2876, 1453, 956, and 842 cm^−1^ were assigned to the C–H stretching and bending [20]. Furthermore, the absorption peaks at 1143, 1103, and 1052 cm^−1^ were detected, owing to the C–O–C stretching and vibration in the PEG. The broad peaks in HP-β-CD FTIR spectra at 3420 cm^−1^ could be assigned to the O–H stretching, and the peaks at 2892 and 1028 cm^−1^ were attributed to C-H stretching and C–O bending vibrations, respectively [21].

The FTIR spectra of MDI shows a broad peak at 2276 cm^−1^ corresponding to the NCO stretching vibration. Furthermore, it can be noted that the FTIR spectra of PU copolymer has all the characteristic peaks of PEG and HP-β-CD, with minor shifts and a reduction in intensity. The absorption band at 2276 cm^−1^ corresponding to NCO stretching vibrations was observed in the FTIR spectra of MDI. The absorption peak due to NCO was absent in the case of the crosslinked PU-PEG copolymer. The broad peak of HP-β-CD at 3300–3500 cm^−1^ resembled stretching of the O-H group. However, the broad peak due to O-H also loses some relative intensity in the case of the crosslinked PU-PEG copolymer. Two different molecular weights, PEG-1500 and PEG-3000, showed almost the same FTIR structure, apart from the O-H peaks between 3300 and 3500 cm^−1^, where PEG-3000 showed a broad peak and PEG-1500 showed a minor characteristic peak, which was attributed to the change in molecular weight of the PEG. It is worth mentioning that new characteristic peaks at 1734, 1550, and 1603 cm^−1^ were observed, suggesting the presence of N-H, C=O, and benzene stretching, respectively, which confirmed the crosslinking of the copolymers using PEG, MDI, and HP-β-CD.

XRD diffractograms were obtained using a double crystal thin film X-ray diffractometer (Bruker D8-Discover). The obtained XRD diffractograms for PEG and PEG-PU nanocomposite are shown in Figure 2b,c, respectively. The pristine PEG demonstrates a crystalline structure, displaying sharp peaks, as shown in Figure 2b. However, after mixing with PU, the composite starts to lose the crystalline properties, with an increase in the amount of PU. As we can see from Figure 2c, PEG and PU with a ratio of 0.2:1 and 0.25:1 have no crystalline properties, while a 0.5:1 and 1:1 ratio shows the presence of crystalline peaks. This suggests that 0.5:1 is the threshold ratio for mixing PEG and PU to enable double phase change properties (solid–liquid) in the nanocomposite.

To investigate the thermosensitive properties of the PEG-PU polymer-based composite, DSC analysis was performed. Based on our previous research and from the DSC analysis, it was found that the mentioned mass ratio range of PEG-PU (0.2:1 to 1:1) is suitable for human body temperature sensing [22]. Figure 3a shows the DSC analysis of pristine PEG and the PEG-PU composites with different mass ratios. Different mass ratios have different melting and crystallization temperatures, depending on the ratio of PEG:PU in the nanocomposite. It can be seen from Figure 3b that the enthalpy of crystallization and crystallization temperature of pristine PEG was the highest. It can also be observed that with an increase in the mass of PU, the crystallization temperature of the PEG-PU composite decreased from 22 °C to 2 °C along with the crystallization enthalpy. The observed change in the crystallization temperature and enthalpy of the composite may have been due to the reduction in mass of PEG in the nanocomposite. Additionally, it has been reported that physical mixing of PU and PEG at a sufficient temperature can lead to a partial cross linking of these two polymers, in such a way that the crystallinity of the PEG decreases [23,24]. This phenomenon appears due to the hydrogen bonding between the electron acceptor hydroxyl hydrogen atom in PEG and the nitrogen atoms present in PU [22,25]. Similarly, the negative heat flow diagram in Figure 3c shows the enthalpy of melting and melting temperature of the composite, which is a direct consequence of the change in mass ratio of PEG and PU, along with the partial crosslinking between these two polymers. It is worth mentioning that the melting temperature of the 0.5:1 PEG-PU composite overlaps the human body temperature range. The enthalpy of melting for this specific ratio (0.5:1) lies in the temperature range of 30 °C to 43 °C, which is an acceptable range for human body temperature monitoring and suggests that our fabricated sensor could exhibit high sensitivity in this temperature range. Figure 3d,e shows the relation of PEG mass fraction ratio with the melting and crystallization temperatures, respectively. Both graphs indicate that with a decrease in the mass of PEG, the crystallization and melting temperature of the composite decreases.

The SEM images of the MWCNT-PU composite are shown in Figure 4. Figure 4a and its magnified image in Figure 4c show the SEM image of 2 wt.% MWCNT in the PU polymer. Since the nanotubes are disjointed, with a noticeable inter-nanotube distance, it can be concluded that tunneling of electrons is required for achieving improved conductivity. It is worth mentioning that this tunneling of electrons can reduce the resistance of PU polymer significantly in a ~100 MΩ range. Figure 4b and its magnified image in Figure 4d show an SEM image of 10 wt.% MWCNT in PU. Here, the interconnected nanotube junctions facilitate electron conduction through the specimen. These interconnected nanotube junctions provide the required pathway for electron tunneling and, consequently, a reduction in the resistivity of the nanocomposite is observed.

While the electrical resistance of pure PU is over 10^12^ Ω, adding a small amount of MWCNT can drastically change the resistance. As observed from Figure 5a, the initial resistance of the PU rapidly decreases to less than 30 MΩ after addition of 2 wt.% MWCNT. This suggests that the presence of MWCNTs allows more electrons to tunnel through the nanocomposite by means of interconnecting and tunneling pathways. Furthermore, a decrease in the resistance of the nanocomposite can be seen with an increase in the conductive filler from 2 wt.% to 8 wt.%. The sharp reduction in the resistance of the nanocomposite at 10 wt.% can be attributed to the formation of interconnected nanotube junctions, due to increased amount of MWCNTs in the nanocomposite [26]. Since the electrons can easily pass through the nanotube junctions, reduced resistance is observed. It has previously been reported that the percolation network and percolation threshold of a nanocomposite depends on the concentration, aspect ratio of nanomaterials, the line width, and line gap of the interdigitated electrode [27]. Here, we have used MWCNT with an aspect ratio ~660 and an interdigitated electrode of line width 0.555 mm and line gap of 0.324 mm. The experimentally observed sharp reduction in resistance (from tens of mega ohm to tens of kilo ohm) can be attributed to the percolation threshold of the nanocomposite. Figure 5b shows the thermoresistive behavior of the PU-MWCNT composite for different loading concentrations (2–10 wt.%) of conductive filler. All the prepared samples with different loading concentrations showed a negative temperature coefficient of resistance. Nanocomposites with 2–8 wt.% of MWCNT showed a noticeable resistance change with increase in temperature from 25 to 50 °C. However, the 10 wt.% nanocomposite showed relatively insignificant changes, due to the high conductivity and low TCR value of MWCNTs. Figure 5c shows the variation of ΔR/R with the temperature. Among the different MWCNT loading concentrations, 8 wt.% nanocomposite showed the maximum change in ΔR/R in the range 25 to 50 °C. The observed results are quite similar to those observed for doped conductive polymers and semiconductors previously reported in the literature [28,29].

Figure 5d shows the thermoresistive behavior of the MWCNT-PEG-PU nanocomposite. Since 8 wt.% MWCNT-PU showed the best thermoresistive behavior, it was selected to further analyze the physically mixed MWCNT-PEG-PU. Various mass ratios from 0.2:1 to 1:1 for PEG-PU were tested to attain the optimized thermosensitive response in the temperature range 25 °C to 50 °C. The expected enhancement in the thermosensitive behavior of the nanocomposite can be seen after addition of different masses of PEG. It is worth mentioning that the 0.5:1 PEG-PU ratio with 8 wt.% MWCNT loading showed the maximum change in resistivity of the nanocomposite, compared to the other tested ratios. The maximum value of ΔR/R for MWCNT-PU was observed to be ~22% at 50 °C, while the same value of ΔR/R for MWCNT-PEG-PU was calculated to be ~80%. The observed change in thermoresistive properties of the nanocomposite was, conclusively, due to an optimized addition of PEG [30,31]. Furthermore, the effect of the molecular weight of PEG (MW = 1500 and 3000) was also analyzed. PEG (M_w_ of 1500 and 3000) was used to make MWCNT-PEG-PU nanocomposite, by means of physical mixing and chemical crosslinking of copolymers. Figure 5e shows the thermoresistive response of synthesized nanocomposites using the different approaches mentioned above, with the error bar observed in the measurement for 7 days. It can be clearly seen that the thermosensitive response of the chemically crosslinked MWCNT-PEG-PU nanocomposite is superior to the physically mixed copolymers for both molecular weights of PEG. It is worth noting that the thermosensitive nature of the PEG with a molecular weight of 1500 is higher compared to the PEG with a molecular weight of 3000. The observed phenomena can be attributed to the fact that a higher molecular weight of PEG is thermally more stable compared to a lower molecular weight [32]. Due to the increased volatility of lower molecular weight PEG in response to heat, it shows a higher sensitivity in the temperature range of 25 to 50 °C. After fabricating the sensor, the flexibility of the sensor was analyzed for 7 days. The sensor was bent with a diameter of 5 mm for 30 continuous cycles, and the change in resistance was noted. The observed changes in the resistance on day 1 and day 7 are depicted in Figure 5f, which were negligible compared to the ΔR/R observed while assessing the thermosensitive properties of the sensor. To perform the temperature measurement and test the feasibility of the sensor for the human body, the sensor was patched on a human body. A comparison of temperature sensing properties of the fabricated sensor and an IR gun on the human body is shown in Figure 5g. Where an error of 0.5 °C was observed while testing the human body temperature. Additionally, the sensor was pasted inside a vacuum chamber oven at 40 °C, and a temperature measurement test was conducted. The observed sensor response (39.7 °C) for the temperature measurement is shown in Figure 5h, on the basis of the thermosensitive behavior in Figure 5e. However, the IR gun failed to measure the temperature of the glass, due to a transparency issue, which further demonstrated the feasibility of the fabricated sensor.

A schematic illustration of the sensing mechanism is shown in Figure 6. The negative temperature coefficient of resistance (NTC) of the nanocomposite is an outcome of solid–liquid double phase of PEG. The melting of PEG results in a volume expansion, which alters the crystallinity of the PEG present in amorphous PU. This fact can be confirmed from Figure 3 [31,33]. As the surrounding temperature approaches the melting point of PEG, it melts and starts to expand. This results in a reduction in the specific volume of PU in the nanocomposite and leads to an increase in the density of MWCNT in the PU phase, which consequently reduces the inter-nanotube distance, as shown in Figure 6a,b [34,35]. As discussed previously, a minor change in the inter-nanotube distance at the nanoscale can lead to a sharp change in the resistance of the material. A similar approach was considered here to elucidate the sensing mechanism. The faint blue parts in Figure 6a,b represent the PU phase, while the yellow part shows the PEG phase. The initial homogeneous dispersion of the MWCNT throughout the nanocomposite can be seen in Figure 6a. However, as the temperature approaches the melting point of PEG, it starts to expand, which compresses the neighboring PU regions, and consequently leads to an increase in the density of MWCNTs in the PU (faint grey), as shown in Figure 6c,d; thus, resulting in a reduction of the overall resistance of the nanocomposite. The NTC behavior of the nanocomposites has been widely recognized by researchers as a consequence of the formation of new conductive chains on exposing the nanocomposites to higher temperature [36,37,38]. Herein, a similar observation was illustrated by using MWCNT-PEG-PU nanocomposites, which demonstrated a NTC behavior, due to formation of new conductive pathways by enhancing the density of MWCNTs in the PU region.

## 4. Conclusions

In summary, we have developed a MWCNT loaded PEG-PU nanocomposite ink that can be printed to develop low-cost and flexible temperature sensors. PEG was used as the thermosensitive material, MWCNT as the thermoresistive conductive filler, and PU as the flexible structural material for loading the PEG by means of physical mixing and chemical crosslinking. The chemically crosslinked MWCNT-PEG-PU nanocomposite showed an improved thermosensitive behavior in the range of 25 °C to 50 °C, compared to physical mixing. The NTC-based sensing mechanism was discussed in detail, and the proposed lightweight, flexible, and sensitive temperature sensor shows promise for a range of wearable applications.

## Figures and Tables

**Figure 1 micromachines-13-00197-f001:**
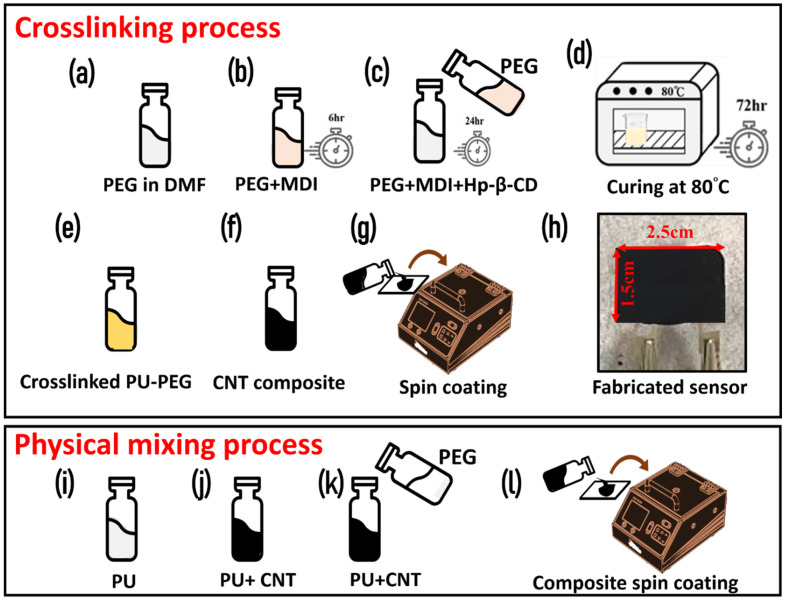
A schematic of the synthesis protocol for (**a**–**d**) crosslinked PEG-PU copolymer; (**e**–**f**) MWCNT-PEG-PU nanocomposite ink; (**g**) spin coating of the ink on IDT electrodes for investigation; (**h**) an image of the fabricated temperature sensor on the IDT electrode; (**i**–**k**) the process to prepare physically mixed MWCNT-PEG-PU nanocomposite; (**l**) spin coating the physically mixed nanocomposite.

**Figure 2 micromachines-13-00197-f002:**
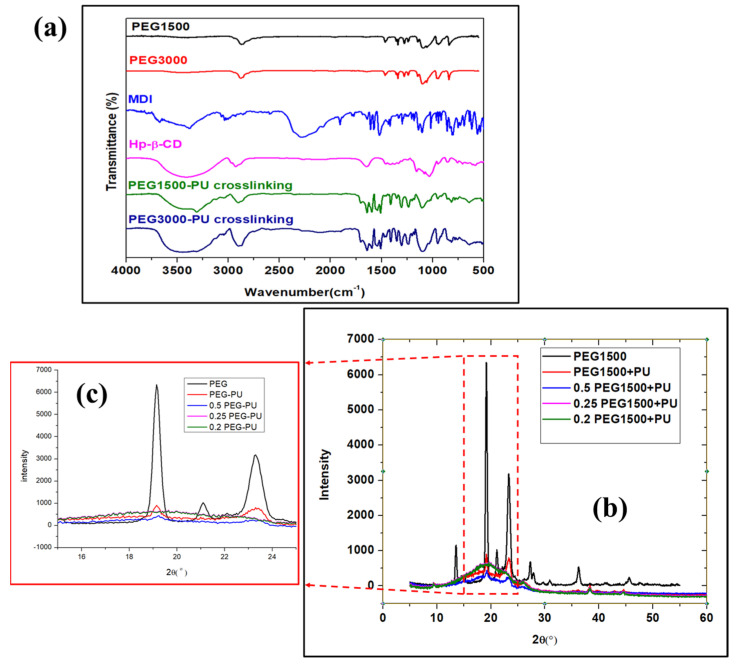
(**a**) FTIR spectra of different polymers and PEG-PU crosslinking copolymer; (**b**) XRD spectra of PEG and physically mixed PEG-PU composite, with a magnified image in (**c**).

**Figure 3 micromachines-13-00197-f003:**
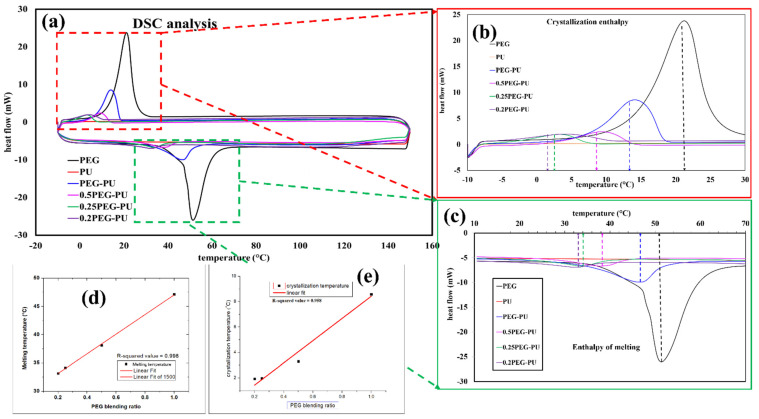
(**a**) DSC analysis of PEG, PU, and the physically mixed composite with different PEG-PU ratios, and a magnified image of (**b**) crystallization enthalpy and crystallization temperature and (**c**) enthalpy of melting and melting temperature; graphical presentation to show the relation between; (**d**) melting temperature and (**e**) crystallization temperature with PEG blending ratio.

**Figure 4 micromachines-13-00197-f004:**
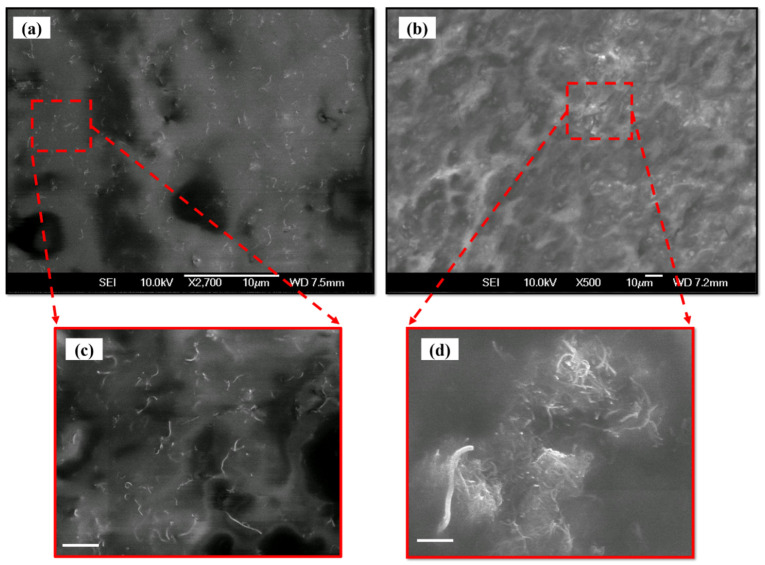
SEM image of MWCNT-PU composite with (**a**) 2 wt.% and (**b**) 10 wt.% loading concentration and their magnified images in figure (**c**,**d**), respectively (scale bar corresponds to 1 µm).

**Figure 5 micromachines-13-00197-f005:**
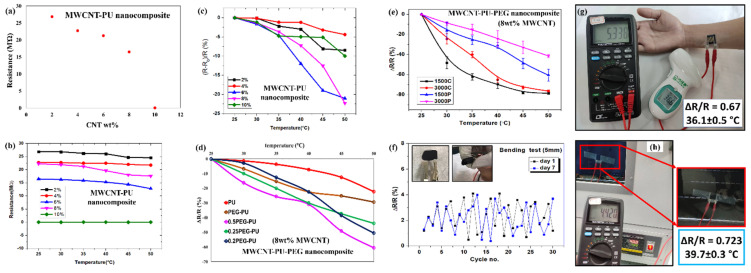
Graphical representation of (**a**) resistance variation with MWCNT loading wt.% in PU; (**b**) change in resistance with temperature of MWCNT-PU nanocomposite; (**c**) variation of ΔR/R with temperature in MWCNT-PU nanocomposite; (**d**) thermoresistive response by means of ΔR/R vs. temperature graph of physically mixed MWCNT-PEG-PU nanocomposites with different ratios; (**e**) a comparison of thermoresistive behavior of physically mixed (PEG1500P and PEG3000P) and chemical crosslinked (PEG1500C and PEG3000C) MWCNT-PEG-PU nanocomposite sensor; (**f**) continuous flexibility test of the fabricated sensor with a bending diameter of 5 mm on day 1 and day 7; temperature measurement test of fabricated sensor (**g**) on a human hand and a comparison with a temperature IR gun (**h**) on oven glass, which was fixed at a temperature of 40 °C, the observed temperature is shown on the bases of ΔR/R.

**Figure 6 micromachines-13-00197-f006:**
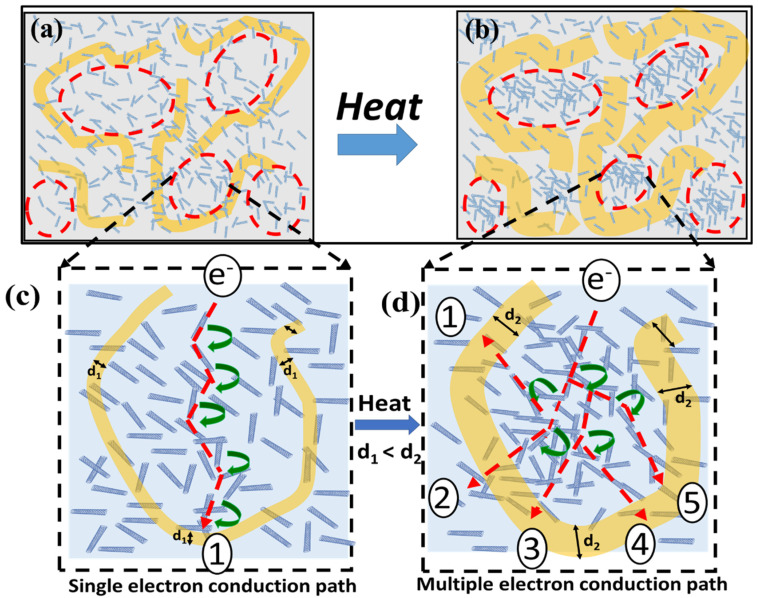
A schematic illustration of the sensing mechanism for the proposed temperature sensor. (**a**) Initial state when PEG (yellow) is in a crystalline state; (**b**) after heating, PEG (yellow) starts to expand, which results in enhancing the MWCNT density in th PU (faint blue) phase of the nanocomposite; magnified image of (**c**) Figure 6a and electron conduction path in the initial state and (**d**) Figure 6b and multiple electron conduction pathways available due to enhanced MWCNT density by means of PEG expansion in the Pu phase.

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
