# Peer review of "Flexible Temperature Sensor Utilizing MWCNT Doped PEG-PU Copolymer Nanocomposites"

_micromachines, 2022, doi:10.3390/mi13020197_

Round 1
Reviewer 1 Report
In this paper, MWCNT loaded PEG-PU nanocomposite have been synthesized and utilized for developing low-cost and flexible temperature sensors. Detailed structural, morphological, thermal and phase transition properties of the nanocomposites were investigated. The aim of the study has some application prospects, but some issues need to be corrected.
- It is better to provide the basis for the value range CNT loading weight percent from 2 wt.% to 10 wt.%.
- The thermosensitive properties of the PEG-PU composites with different mass ratio and the thermoresistive behaviour of the PU-MWCNT composite for different loading concentrations were investigated. Why is there no analysis and comparison of PEG-PU composites for MWCNT loading concentrations?
- It is better to aid a schematic process for physical mixing.
- This suggests that 0.5:1 is the threshold ratio for mixing PEG and PU to enable double phase change properties (solid-liquid) in the nanocomposite. Why mass ratios from 0.2:1 to 1:1 for PEG-PU have been tested to attain the optimized thermosensitive response?
- The quality of figures are low, please replace its with high-quality pictures.
- Are the effects of diameter and length of MWCNT considered? (Please refer to these papers: International Journal of Heat and Mass Transfer 152 (2020) 119565; Carbon, 2021, 178, 391-412.) And it is better to uniform parameters unit.
- There is no measurement of flexibility,stability and repeatability.( Please refer to this paper: Sensors & Actuators: B. Chemical 305 (2020) 127445)
- As wearable and flexible device, will the addition of MWCNT be toxic? Has a temperature test experiment and temperature measurement error were investigated?How to prove that it can be used in practical applications
- It is better to compare the PEG-PU-MWCNT sensor with traditional temperature sensor.
Author Response
Dear Reviewer:
please find the reply letter as attached file for the details.
Best Regards,
Cheng-Hsin Chuang Ph.D. Professor/Chair
Institute of Medical Science and Technology
National Sun Yat-sen University
70 Lienhai Rd., Kaohsiung 80424, Taiwan.
E-mail:chchuang@imst.nsysu.edu.tw Tel: +886-7-5252000 ext 5785
Mobile: 0937929830
Website:https://manstlab.wixsite.com/manstlab/

Reviewer 2 Report
General comments
The manuscript presents a novel path for the realization of the low-cost flexible temperature sensors based on ternary nanocomposites consisting of MWCNT-PEG-PU, where the PEG-PU components were either physically mixed or chemically crosslinked. FTIR, DSC, XRD SEM analyses were used for material characterization followed by evaluation of the electrical resistance as function of MWCNT loading, PEG-PU composition and temperature. The percolation threshold was also found to be located between 8 and 10 MWCNT wt%. The highest temperature sensitivity was found for MWCNT loading equal to 8 wt% in MWCNT with PEG1500C/PU=1/2 . The authors have also presented an intuitive model to describe the negative temperature coefficient of the resistance as a function of temperature. The manuscript can be published after solving more issues described below and also indicated as comments inside their manuscript.
Specific comments
- The highest sensitivity of the temperature sensor is obtained for 8 wt% in MWCNT in the ternary nanocomposite with PEG1500C/PU=1/2 and this is explained by volatility of PEG1500C. The question is how reliable is the sensor under these conditions, when a component like PEG is volatile?. Please comment on long terms stability of this sensor.
- The percolation threshold of the MWCNT in the PU polymer should be better commented in the frame work of the sensor geometry, as follows. In a recent paper, Sensors 2021, 21, 1435. https://doi.org/10.3390/s21041435, it was demonstrated that the percolation threshold depends also on the spacing size between the metal digits of the interdigitated metal geometry used. So, this percolation threshold which you mentioned should be assigned to your interdigit spacing! You should comment this aspect in your paper, based on the reference paper.
- The effect of the PU component on the crystallization and melting temperature of the composite is rather confusedly described. Decreasing the fraction of PEG by adding more PU in the nanocomposite, the crystallization and melting temperatures of nanocomposite are decreasing, therefore more PU does not resist to crystallization/melting, but on the contrary.
- There is no explicit comment in the text related to Fig. 1 a-d.
- The Fig. 5 and its legend are not self-explanatory together. More details are presented in your manuscript where comments are inserted.
- The model is intuitive, but it should be based on prior-art material results. Citations are needed for the PEG volume expansion as a function of phase transition to liquid state. Similarly, the contraction of PU as function of temperature and PEG expansion should be based on citations.
- More typo details are marked in your attached manuscript.

Author Response
Dear Reviewer:
Please find the reply letter as attached file for the details.
Thanks!
Best Regards,Cheng-Hsin Chuang Ph.D. Professor/Chair
Institute of Medical Science and Technology
National Sun Yat-sen University
70 Lienhai Rd., Kaohsiung 80424, Taiwan.
E-mail:chchuang@imst.nsysu.edu.tw Tel: +886-7-5252000 ext 5785
Mobile: 0937929830
Website:https://manstlab.wixsite.com/manstlab/

Reviewer 3 Report
The authors presented a study of a copolymer nanocomposite based on MWCNT-doped PEG-PU for use as a flexible temperature sensor. In order for it to be acceptable for publication, the authors need to address some concerns:
- In Figure 2, the font size of labels a, b, and c are too big please re-adjust it based on the journal requirements. It is not to stretch the figure (FTIR spectra) to make sure it has the same scale. This also applies to Figure 4.
- The authors presented a scheme on how their temperature sensors work. It is better to provide experimental evidence of the said scheme such as SEM images before and after heating or other characterization techniques or do a theoretical modeling study.
Author Response
Dear Reviewer:
Please find the reply letter as attached file for the details.
Thank you very much.
Best Regards,Cheng-Hsin Chuang Ph.D. Professor/Chair
Institute of Medical Science and Technology
National Sun Yat-sen University
70 Lienhai Rd., Kaohsiung 80424, Taiwan.
E-mail:chchuang@imst.nsysu.edu.tw Tel: +886-7-5252000 ext 5785
Mobile: 0937929830
Website:https://manstlab.wixsite.com/manstlab/

Round 2
Reviewer 1 Report
I recommend accept for this paper.
Reviewer 2 Report
The authors followed the reviewer's recommendation. There are still some minor typo in the manuscript, which should be corrected.
I am in favor of publishing the manuscript.
Reviewer 3 Report
The authors have sufficiently addressed the comments/suggestions made by the reviewers.